# Universal signatures of Majorana zero modes in critical Kitaev chains

Nicolas Laflorencie

*Laboratoire de Physique Théorique, Université de Toulouse, CNRS, UPS, France*

Many topological or critical aspects of the Kitaev chain are well known, with several classic results. In contrast, the study of the critical behavior of the strong Majorana zero modes (MZM) has been overlooked. Here we introduce two topological markers which, surprisingly, exhibit non-trivial signatures over the entire (1+1) Ising critical line. We first analytically compute the MZM fidelity $\mathcal{F}_{\mathrm{MZM}}$–a measure of the MZM mapping between parity sectors. It takes a universal value along the (1+1) Ising critical line, $\mathcal{F}_{\mathrm{MZM}} = \sqrt{8}/\pi$, independent of the energy. We also obtain an exact analytical result for the critical MZM occupation number $\mathcal{N}_{\mathrm{MZM}}$ which depends on the Catalan's constant $\mathcal{G} \approx 0.91596559$, for both the ground-state ($\mathcal{N}_{\mathrm{MZM}} = 1/2 - 4\mathcal{G}/\pi^2 \approx 0.12877$) and the first excited state ($\mathcal{N}_{\mathrm{MZM}} = 1/2 + (8 - 4\mathcal{G})/\pi^2 \approx 0.93934$). We further compute finite-size corrections which identically vanish for the special ratio $\Delta/t = \sqrt{2} - 1$ between pairing and hopping in the critical Kitaev chain.

## I. INTRODUCTION

*Generalities*— Much attention has recently turned towards emerging Majorana bound states in certain condensed matter systems [1–7]. One of the simplest toy-model hosting such a fascinating physics is an exactly solvable quantum chain model, solved in the magnetic language quite some time ago by Lieb, Schulz, Mattis [8] and Pfeuty [9]. Nevertheless, a decisive step was later taken thanks to the seminal work of Kitaev [10] who realized that non-trivial topological properties could emerge in such a simple quantum model when rephrased in fermionic language, nowadays referred to as the Kitaev chain

$$\mathcal{H}_{\mathrm{K}} = -\sum_j \left( [t_j c_j^\dagger c_{j+1} + \Delta_j c_j^\dagger c_{j+1}^\dagger + \mathrm{h.c.}] - \mu_j c_j^\dagger c_j \right) \quad (1.1)$$

which describes a non-interacting p-wave superconducting wire with hopping $t_j$, pairing $\Delta_j$ and potential $\mu_j$. Equivalently it represents the original [8] spin chain XY Hamiltonian

$$\mathcal{H}_{\mathrm{K}} = -\sum_j \left( X_j \sigma_j^x \sigma_{j+1}^x + Y_j \sigma_j^y \sigma_{j+1}^y + h_j \sigma_j^z \right), \quad (1.2)$$

with couplings $X_j = \frac{t_j + \Delta_j}{2}$, $Y_j = \frac{t_j - \Delta_j}{2}$, and fields $h_j = \frac{\mu_j}{2}$.

The total number of fermions $N_{\mathrm{f}} = \sum_j c_j^\dagger c_j$ ($\sum_j (\sigma_j^z + 1)/2$ in the spin language) is not fixed (unless $\Delta_j = X_j - Y_j = 0$), but its parity is conserved: $\mathbb{P} = (-1)^{N_{\mathrm{f}}} = \prod_j \sigma_j^z$ has eigenvalues $p = \pm 1$ and commutes with $\mathcal{H}_{\mathrm{K}}$. This yields a global $\mathbb{Z}_2$ symmetry to the problem and one can therefore group the eigenstates in two distinct parity sectors. The spontaneous breaking of $\mathbb{Z}_2$ symmetry may occur in the thermodynamic limit, associated with magnetic order $\langle \sigma^x \rangle \neq 0$. This standard long-range order takes a non-local form in fermionic language, with "topological" unpaired zero-energy Majorana edge states localized at the boundaries [10].

*Strong Majorana zero modes*— The idea of *strong* Majorana zero-mode (MZM), popularized by Fendley [11, 12] and collaborators [13–15], goes beyond low energy [16, 17] as the *whole* many-body spectrum is involved, see inset (i) of Fig. 1. A strong MZM operator $\psi$ has the following key properties (assume a $L$ site open chain): (1) it commutes with the Hamiltonian (at least for $L \gg 1$) $[\mathcal{H}, \psi] \to 0$; (2) it anti-commutes with the discrete symmetry (here the parity $\{\mathbb{P}, \psi\} = 0$); and (3) it is normalizable $\psi^\dagger \psi = \psi^2 = 1$.

Introducing two Majorana fermions at each site $a_j = c_j^\dagger + c_j$ and $b_j = \mathrm{i}(c_j^\dagger - c_j)$, the above XY-Kitaev chain model rewrites

$$\mathcal{H}_{\mathrm{K}} = \mathrm{i} \sum_j \left( X_j b_j a_{j+1} - Y_j a_j b_{j+1} + h_j a_j b_j \right). \quad (1.3)$$

Assuming site-independent couplings ($h/X \geq 0$, $X > Y \geq 0$), one can construct two such strong MZM operators

$$\psi_a = \frac{1}{N_a} \sum_{j=1}^L \Theta_j^a a_j \quad \text{and} \quad \psi_b = \frac{1}{N_b} \sum_{j=1}^L \Theta_j^b b_{L+1-j}, \quad (1.4)$$

which both commute with $\mathcal{H}_{\mathrm{K}}$ in the $L \to \infty$ limit under the simple condition $h < X + Y$ [18], see Fig. 1 (a). They decay exponentially away from the left and right boundaries

$$\left| \Theta_j^{a,b} \right| \propto \exp\left( -\frac{j-1}{\xi_{\mathrm{zm}}} \right). \quad (1.5)$$

The MZM localization length diverges if $X + Y - h \to 0^+$, following $\xi_{\mathrm{zm}} \approx \frac{X-Y}{X+Y-h}$ [18], thus ensuring their normalization $\psi_{a,b}^2 = 1$, with a finite norm

$$N_{a,b} = \sqrt{\sum_{j=1}^L \left| \Theta_j^{a,b} \right|^2} < \infty \quad \text{if} \quad h < X + Y. \quad (1.6)$$

The strong character of the MZMs in Eq. (1.4) is rooted in the anti-commutation property $\{\mathbb{P}, \psi_{a,b}\} = 0$ [19]. Indeed, when combined with $[\mathcal{H}_{\mathrm{K}}, \psi_{a,b}] = 0$, we can build the following non-local (bilocalized at both edges) Dirac fermion operator

$$\Psi^\dagger = \frac{1}{2} \left( \psi_a - \mathrm{i} \psi_b \right) \quad (1.7)$$

which creates a zero-energy fermion, and provides a mapping between the two parity sectors *for all states*. In the thermodynamic limit, the topological regime is characterized by

$$\langle n_p | \Psi^\dagger \Psi | n_p \rangle = \begin{cases} 1 & \Rightarrow \quad \Psi | n_p \rangle = | n_{-p} \rangle \\ 0 & \Rightarrow \quad \Psi^\dagger | n_p \rangle = | n_{-p} \rangle, \end{cases} \quad (1.8)$$

for any many-body eigenstate $| n_p \rangle$ of parity $p = \pm 1$, where all the energies are pairwise degenerate $E_n^+ = E_n^-$.

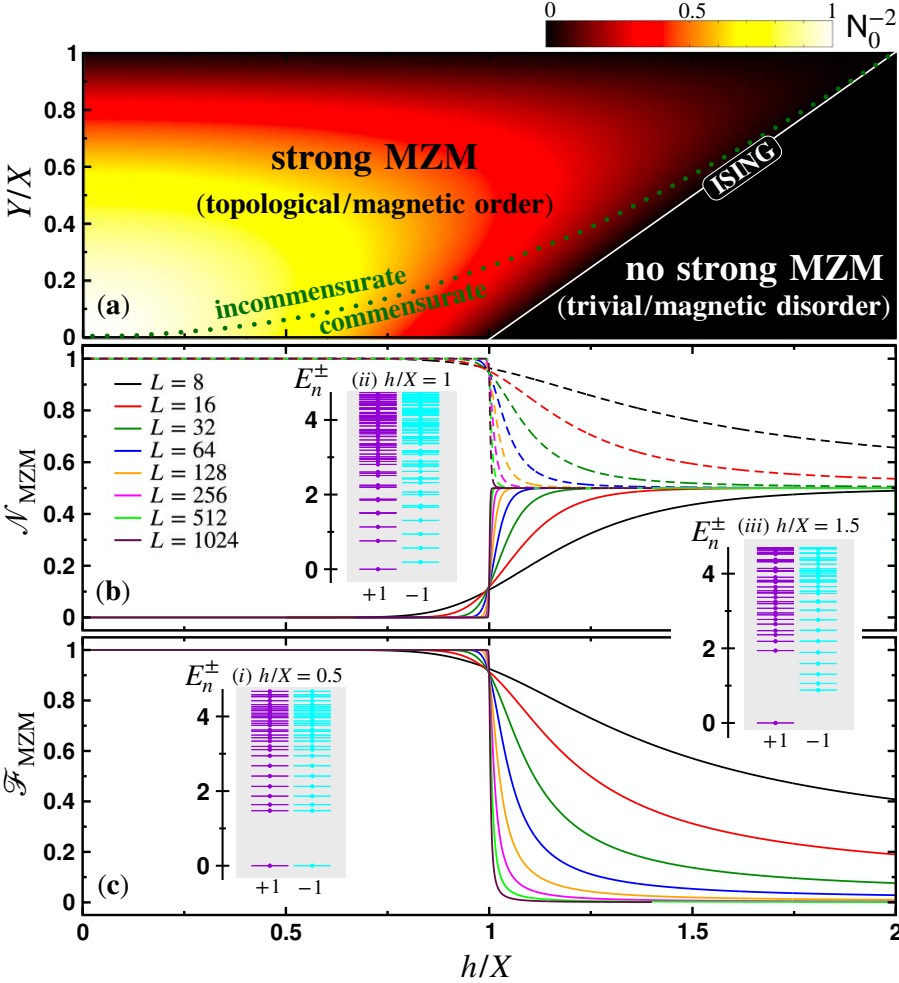

FIG. 1. **(a)** Phase diagram of the clean Kitaev-XY chain model Eqs. (1.1)-(1.2). The Ising transition line $h = X + Y$ (white line) separates topological and trivial regimes. The color map describes the inverse norm squared of the MZM Eq. (1.6) with $N_0 = N_{a,b}$. The commensurate-incommensurate line inside the topological regime is shown by the dotted line. Panels **(b-c)** are scans along the $Y = 0$ line of the phase diagram (corresponding to the transverse-field Ising model) showing exact diagonalization (ED) results for open chains of various sizes $L$, as indicated on the plot. **(b)** The MZM occupation number $\mathcal{N}_{\mathrm{MZM}} = \langle \Psi^\dagger \Psi \rangle$, computed in the ground-state (full lines) and the first excited state (dotted lines), takes asymptotic values 0 or 1 in the ordered phase ($h < X$), 0.5 on the disordered side ($h > X$), and non-trivial universal values at the quantum critical point (QCP) $\mathcal{N}_{\mathrm{MZM}} = 1/2 - 4\mathcal{G}/\pi^2 \approx 0.1287731$ for the GS and $\mathcal{N}_{\mathrm{MZM}} = 1/2 + (8 - 4\mathcal{G})/\pi^2 \approx 0.939342$ for the first excited state, where $\mathcal{G} \approx 0.915966$ is the Catalan's constant (see text). **(c)** The MZM fidelity $\mathcal{F}_{\mathrm{MZM}}$ Eq. (1.9) is shown across the transition for the same system sizes. It takes asymptotic values 1 or 0 in the two phases, and a non-trivial universal number $\mathcal{F}_{\mathrm{MZM}}^{\mathrm{critical}} = \sqrt{8}/\pi \approx 0.900316$ at the Ising QCP. The three insets *(i-iii)* show the bottom of the many-body spectrum for $L = 16$ in the three regimes, resolved in term of the parity quantum number $p = \pm 1$.

Fig. 1 summarizes these classic results for the clean Kitaev-XY chain, where two topological markers are shown: the zero-mode occupation number $\mathcal{N}_{\mathrm{MZM}} = \langle \Psi^\dagger \Psi \rangle$ for both the ground-state (GS) and the first excited state (FES), and the MZM fidelity, defined by

$$\mathcal{F}_{\mathrm{MZM}}^{(n)} = \frac{1}{2} \langle n_{-p} | \psi_a \pm i\psi_b | n_p \rangle, \quad (1.9)$$

which quantifies the connection between the two parity sectors via the MZM mapping Eq. (1.8).

*Main results and paper outline*— In this work we present analytical calculations of these two topological quantities. Along the Ising quantum critical line $h = X + Y$, de-

spite its non-topological nature, we surprisingly find a finite universal value for the fidelity $\mathcal{F}_{\mathrm{MZM}}^{(n)} = \sqrt{8}/\pi$, $\forall n$, as well as for the two lowest MZM occupation numbers, $\mathcal{N}_{\mathrm{MZM}} = 1/2 - 4\mathcal{G}/\pi^2 \approx 0.1287731$ for the ground-state, and $\mathcal{N}_{\mathrm{MZM}} = 1/2 + (8 - 4\mathcal{G})/\pi^2 \approx 0.939342$ for the first excited state, where $\mathcal{G} \approx 0.915966$ is the Catalan's constant [20]. In the rest of the paper, we present the analytical derivations, that we then carefully check numerically using large scale exact diagonalization up to $L \sim 10^4$ lattice sites. We also prove analytically the universality of our results along the Ising quantum critical line of the clean Kitaev-XY chain model. Interestingly, we identify a special critical point at $h = X + Y = \sqrt{2}$ where finite corrections vanish completely.

## II. MAJORANA ZERO MODE FIDELITY IN THE CLEAN KITAEV CHAIN

### A. Relation between the fidelity and the parity gap

We first consider a uniform chain of $L$ sites with free ends. The parity gap is the energy difference within the even-odd doublet, defined by

$$\Delta_{\text{parity}}^{(n)} = \mathcal{S}_n\Big(\langle n_- |\mathcal{H}| n_- \rangle - \langle n_+ |\mathcal{H}| n_+ \rangle\Big), \quad (2.1)$$

where the sign $\mathcal{S}_n$ ensures that $\Delta_{\text{parity}}^{(n)}$ is positive. In fact, this term has a physical meaning: it reflects the sign of the end-to-end correlations between the boundary spins [21].

The even-odd mapping is rigorously given by

$$\frac{\psi_a + \mathrm{i}\mathcal{S}_n\mathbb{P}\psi_b}{2} | n_p \rangle = \mathcal{F}_{\text{MZM}}^{(n)} | n_{-p} \rangle - \sum_{n' \neq n} \alpha_{n'} | n'_{-p} \rangle, \quad (2.2)$$

yielding the following expression for the parity gap

$$\Delta_{\text{parity}}^{(n)} = \frac{\mathcal{S}_n \langle n_- | [\mathcal{H}, \psi_a] | n_+ \rangle - \mathrm{i}\langle n_- | [\mathcal{H}, \psi_b] | n_+ \rangle}{2\mathcal{F}_{\text{MZM}}^{(n)}} \quad (2.3)$$

where one recognizes the commutators between the Hamiltonian and the MZMs $[\mathcal{H}, \psi_{a,b}]$. This expression will be used below to get exact forms of the MZM fidelity.

### B. The transverse field Ising chain: analytical results

The special point where hopping equals pairing in the fermionic Kitaev chain Eq. (1.1) corresponds to the paradigmatic transverse field Ising (TFI) chain model which is a cornerstone of quantum statistical physics [9, 22]:

$$\begin{aligned} \mathcal{H}_{\text{TFI}} &= -\sum_j \Big(X\sigma_j^x \sigma_{j+1}^x + h\sigma_j^z\Big) \\ &= \mathrm{i}\sum_j \big(Xb_j a_{j+1} + h_j a_j b_j\big). \end{aligned} \quad (2.4)$$

Its relative simple form will allow us to provide some details on the analytical calculation of $\mathcal{F}_{\text{MZM}}^{(n)}$. The central quantities are the MZM commutators with $\mathcal{H}_{\text{TFI}}$, here given by

$$[\mathcal{H}_{\text{TFI}}, \psi_a] = -\frac{2\mathrm{i}X}{N_a}\Gamma^L b_L, \quad [\mathcal{H}_{\text{TFI}}, \psi_b] = \frac{2\mathrm{i}X}{N_b}\Gamma^L a_1, \quad (2.5)$$

where $\Gamma = h/X$ is the control parameter for the transition. If one expresses the boundary operators in the spin language $a_1 = \sigma_1^x$ and $b_L = -\mathrm{i}\mathbb{P}\sigma_L^x$, the parity gap Eq. (2.3) can be rewritten as follows

$$\Delta_{\text{parity}}^{(n)} = \frac{X}{\mathcal{F}_{\text{MZM}}^{(n)}}\Big(\frac{m_1^s}{N_b} + \frac{m_L^s}{N_a}\Big)\Gamma^L. \quad (2.6)$$

Here we have introduced the surface magnetization [23, 24] $m_{1,L}^s = \big|\langle n_- |\sigma_{1,L}^x| n_+ \rangle\big|$ whose magnitude is independent of

the state $n$ in the free-fermion case, so is the parity gap (see A+). Moreover, the MZM norms Eq. (1.6), both equal for clean chains, are given by

$$N_{a,b} = \sqrt{\frac{1 - \Gamma^{2L}}{1 - \Gamma^2}} \equiv \mathsf{N}_0, \quad (2.7)$$

which hence leads to the following expression for the (energy-independent) MZM fidelity of finite clean TFI chains

$$\mathcal{F}_{\text{MZM}}(L) = \frac{2Xm^s}{\Delta_0 \mathsf{N}_0}\Gamma^L, \quad (2.8)$$

where $\Delta_0$ is the lowest energy gap, $\mathsf{N}_0 = N_{a,b}$ is the MZM norm in Eq. (2.7), and $m^s = m_{1,L}^s$ is the surface magnetization, similar at each boundary of a clean chain. Eq. (2.8) turns out to be valid across the entire phase diagram for finite chains, as we discuss now for the three physical regimes.

#### 1. Disordered phase $\Gamma > 1$

The trivial (disordered) phase for $h > J$ has a finite energy gap above the GS, $\Delta_0 = 2X(\Gamma - 1)$, a diverging MZM norm $\mathsf{N}_0 \sim \Gamma^{L-1}$, and a power-law vanishing surface magnetization $m^s \sim L^{-3/2}$ [9]. This, when put together, leads to the following power-law decay for the MZM fidelity in the topologically trivial phase

$$\mathcal{F}_{\text{MZM}}^{\text{trivial}}(L) \sim L^{-3/2}, \quad (2.9)$$

which asymptotically matches the expected vanishing in the disordered phase. However, the relatively slow algebraic decay is not a trivial result, in a regime where one would have naively expected a faster exponential decay, see below Sec. II C for a numerical check of this result.

#### 2. Ordered regime $\Gamma < 1$

In contrast, in the topological (ordered) regime one can show that $m^s = 1/\mathsf{N}_0 = \sqrt{1 - \Gamma^2}$. Moreover, the gap is exponentially small [9, 25, 26] $\Delta_0 = 2X(1 - \Gamma^2)\Gamma^L + \mathcal{O}(\Gamma^{2L})$, which yields

$$\mathcal{F}_{\text{MZM}}^{\text{topo}}(L) = 1 - \mathcal{O}(\Gamma^{2L}), \quad (2.10)$$

as expected from the strong MZM definition. The finite-size corrections $\sim \exp(-2L/\xi_{\text{zm}})$ are exponentially small, controlled by the MZM localization length

$$\xi_{\text{zm}} = \frac{1}{\ln(1/\Gamma)}. \quad (2.11)$$

#### 3. Quantum critical point $\Gamma = 1$

Perhaps the most remarkable result concerns the quantum critical point (QCP) itself. Indeed, for $\Gamma = 1$ both the finite-size gap and the surface magnetization vanish, $\Delta_0 \sim 1/L$ and

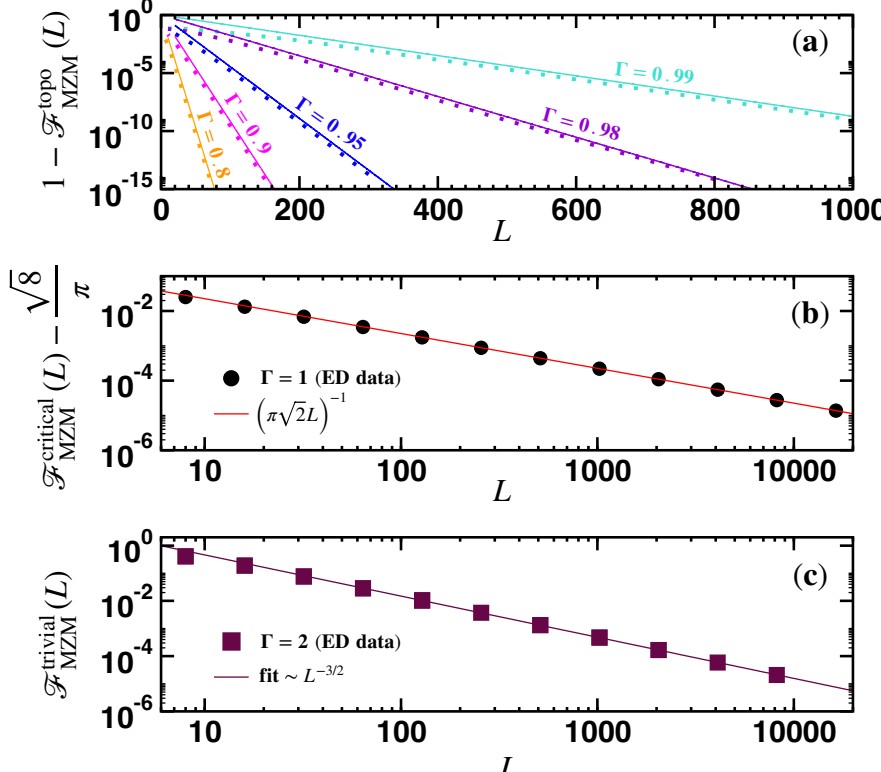

FIG. 2. Finite-size scaling of the MZM fidelity for the TFI chain model across the three different regimes. Comparison between analytical predictions (full lines) and exact diagonalization (ED) results (symbols) obtained from free-fermion calculations, see Sec. II C. **(a)** In the topological regime, one verifies the very fast convergence $\mathcal{F}_{\text{MZM}}^{\text{topo}} \to 1$ with $L$, following Eq. (2.10), displayed for various values of $\Gamma = h/X < 1$: lines are $\Gamma^{2L}$ and symbols show ED data. **(b)** At criticality $\Gamma = 1$, the $1/L$ convergence towards $\sqrt{8}/\pi$ shows perfect agreement between ED data (symbols) and the analytical prediction Eq. (2.14) (red line). **(c)** In the disordered regime $\Gamma > 1$ the expected analytical scaling Eq. (2.9) provides a perfect description of ED data.

$m^s \sim 1/\sqrt{L}$, while the MZM norm $\mathsf{N}_0 = \sqrt{L}$. This implies for Eq. (2.8) a finite critical fidelity, a results already visible in Fig. 1 (c) where a finite-size crossing occurs. One can be more precise, using the critical scaling of the gap [25]

$$\Delta_0(L) = X k_{\min}(L), \qquad (2.12)$$

where $k_{\min}$ is the lowest mode (determined from the boundary conditions) that one can write

$$k_{\min}(L) = \frac{\pi}{L + \ell_{\text{eff}}}. \qquad (2.13)$$

OBC imply [25] $\sin k(L+1) = -\sin kL$, which yields $\ell_{\text{eff}} = \frac{1}{2}$.

The critical scaling of the surface magnetization can be obtained when the QCP is approached from the ordered side $\Gamma \to 1^-$, where $m^s \approx \sqrt{2/\xi_{\text{zm}}}$ for large enough MZM localization length. At the QCP where $\xi_{\text{zm}}$ is formally infinite, it is standard to replace this length scale by the lattice size $L$. Interestingly we numerically find that the correct length scale to use is precisely the one which enters in the wave vector $k_{\min}$, i.e. $L + \ell_{\text{eff}} = L + 1/2$ (see Appendix B). When plugged into Eq. (2.8), we then arrive at a rather simple expression for the critical fidelity

$$\mathcal{F}_{\text{MZM}}^{\text{critical}}(L) = \frac{\sqrt{8}}{\pi} + \left(\pi\sqrt{2}L\right)^{-1} + O\left(L^{-2}\right), \qquad (2.14)$$

which converges to $\sqrt{8}/\pi \approx 0.90032$, with finite-size algebraic corrections $\sim L^{-1}$.

### C. Numerical results

These analytical results can be checked against exact diagonalization simulations of the free fermionic Hamiltonian that we perform for open chains of large length $L$. Fig. 2 shows such exact finite-size computations of the MZM fidelity for the three different regimes where we nicely observe that the analytical predictions for the trivial regime Eq. (2.9), the topological phase Eq. (2.10), and at criticality Eq. (2.14), all perfectly agrees with the exact numerics. In Appendix A we provide some details about the free-fermion exact diagonalization method, and how the MZM operators and fidelity are obtained.

## III. UNIVERSAL FIDELITY AT ISING CRITICALITY

### A. Analytical results

We now discuss how universal this result is: we repeat a similar calculation for the more generic Kitaev-XY model Eq. (1.3), moving away from the $t = \Delta$ point along the Ising quantum critical line at $h = X + Y$, see Fig. 1 (a).

#### 1. Derivation of the critical fidelity

We restart from the definition of the parity gap Eq. (2.3). The first important quantity to compute are the commutators $[\mathcal{H}_K, \psi_{a,b}]$, which are straightforward to get at criticality

$$[\mathcal{H}, \psi_a] = \frac{-2iX^2}{N_a(X - Y)} b_L \; ; \; [\mathcal{H}, \psi_b] = \frac{2iX^2}{N_b(X - Y)} a_1, \quad (3.1)$$

where the norm is

$$N_{a,b} = \mathsf{N}_0 = \frac{X}{X - Y} \sqrt{L - \ell'_{\text{eff}}} \quad (3.2)$$

with (see Appendix C 1 b)

$$\ell'_{\text{eff}} = \frac{y(2 + y)}{1 - y^2}, \quad (y = Y/X). \quad (3.3)$$

Injecting this expression in Eq. (2.3) we arrive at

$$\mathcal{F}_{\text{MZM}}^{\text{critical}}(L) = \frac{2Xm^s}{\Delta_0 \sqrt{L - \ell'_{\text{eff}}}}, \quad (3.4)$$

where as before $\Delta_0$ is the lowest gap, and $m^s$ the surface magnetization, both evaluated along the Ising quantum critical line. There, one can use [27]

$$\Delta_0(L) \approx (X - Y)k_{\min} = \frac{(X - Y)\pi}{L + \ell_{\text{eff}}} \quad (3.5)$$

and similar to the TFI case (see Appendix B 3)

$$m^s(L) \approx \frac{X - Y}{X} \sqrt{\frac{2}{L + \ell_{\text{eff}}}}, \quad (3.6)$$

which gives after replacing in Eq. (3.4)

$$\mathcal{F}_{\text{MZM}}^{\text{critical}}(L) = \frac{\sqrt{8}}{\pi} \sqrt{\frac{L + \ell_{\text{eff}}}{L - \ell'_{\text{eff}}}} \quad (3.7)$$

$$= \frac{\sqrt{8}}{\pi} + \frac{\sqrt{2}\left(\ell_{\text{eff}} + \ell'_{\text{eff}}\right)}{\pi} L^{-1} + O\left(L^{-2}\right). \quad (3.8)$$

#### 2. Finite-size corrections towards the universal value

Here a few comments are in order. First we find that the asymptotic critical value of the MZM fidelity $\sqrt{8}/\pi$ turns out

to be universal along the Ising quantum critical line. Then, the leading finite size corrections are proportional to $L^{-1}$, with a prefactor which involves $\ell'_{\text{eff}}$ and $\ell_{\text{eff}}$. We can check that the TFI result Eq. (2.14) is perfectly recovered using $\ell_{\text{eff}} = 1/2$ and $\ell'_{\text{eff}} = 0$ at $Y = 0$. For finite $Y$ however, it is more cumbersome to explicitly compute $\ell'_{\text{eff}}$. Interestingly, Ref. [27] gave the following ansatz for this effective length shift

$$\ell_{\text{eff}} = \frac{1}{2} - \frac{y(3 + y)}{1 - y^2}. \quad (3.9)$$

Plugging this ansatz in Eq. (3.7) we see that the finite size corrections change sign and cancel out exactly for $y + 1 = \sqrt{2}$, i.e. when $h_c/X = \sqrt{2}$.

### B. Numerical results

#### 1. Ising transitions

The MZM fidelity is obtained from ED calculations, for various chains with OBC, typically ranging from $L = 16$ to $L = 16384$ sites. Fig. 3 (a-b) show $\mathcal{F}_{\text{MZM}}$ for various cuts in the phase diagram of the Kitaev-XY Hamiltonian, see Fig. 1 (a). Exactly as was observed for the TFI chain in Fig. 1 (c), here also the transition between trivial ($\mathcal{F}_{\text{MZM}} \to 0$) and topological ($\mathcal{F}_{\text{MZM}} \to 1$) regimes is signalled by a finite-size crossing at $\mathcal{F}_{\text{MZM}} = \sqrt{8}/\pi$, thus confirming this universal number along the entire Ising critical line of the XY-Kitaev model.

#### 2. Finite-size convergence

Fig. 3 (c-d) show how the critical fidelity approaches this universal value. In panel (c) one sees a clear convergence with $L$ to $\sqrt{8}/\pi$, which features distinct finite-size effects as a function of $y = Y/X$. Indeed, as expected from the previous part, the finite-size corrections change sign for $y = \sqrt{2}-1$. This is better seen in panel (d) in a log scale where the numerical data perfectly compare to the analytical expression Eq. (3.7). In terms of the Kitaev chain parameters, this occurs for a particular ratio between pairing and hopping $\Delta/t = \sqrt{2} - 1$.

#### 3. Effective length scales

The vanishing of finite-size corrections is due to the fact that the effective length scales emerging in the norm $\ell'_{\text{eff}}$ and from the gap $\ell_{\text{eff}}$ cancel each other in Eq. (3.7), provided that $\ell_{\text{eff}} + \ell'_{\text{eff}} = 0$. While the analytical expression for $\ell'_{\text{eff}}$ in Eq. (3.3) can be derived exactly from the normalization of the zero mode operator (see Appendix C 1 b), we have tested the ansatz Eq. (3.9) proposed by Campostrini and co-workers in Ref. [27], which indeed perfectly matches our ED data. This is shown in Fig. 3 (e) where $1/2 - \ell_{\text{eff}} = y(3+y)/(1-y^2)$ is plotted together with ED estimates extracted from the gap Eq. (3.5) and the surface magnetization Eq. (3.6). The agreement is excellent and gets clearly better at large sizes when $y \to 0$.

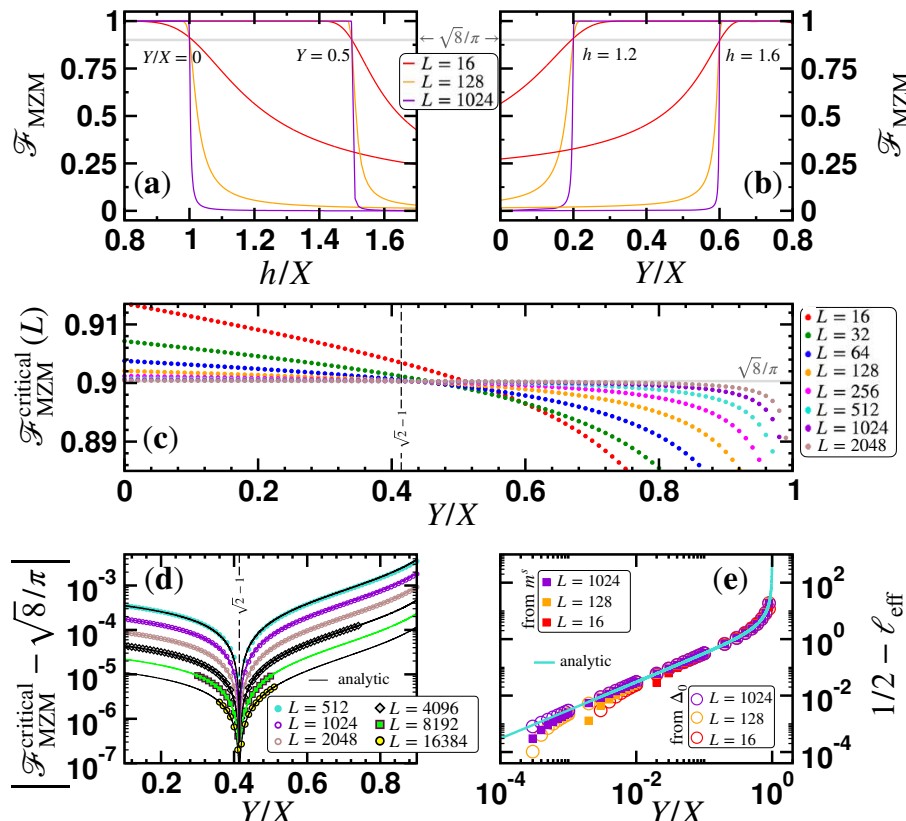

FIG. 3. Finite-size scaling of the MZM fidelity for the Kitaev-XY model Eq. (1.3). Free-fermion ED results are shown as a function of $Y/X$ and $h/X$, see the phase diagram in Fig. 1 (a). (a-b) $\mathcal{F}_{\text{MZM}}$ across the transition along two horizontal cuts (a) or vertical cuts (b) for 3 representative system sizes: one sees the curves crossing at $\sqrt{8}/\pi$ in all cases. (c) $\mathcal{F}_{\text{MZM}}$ is shown along the critical line $h = X + Y$ as a function of $Y$ for various $L$. One sees the finite-size convergence towards $\sqrt{8}/\pi$ with a vanishing of finite-size corrections at $Y/X = \sqrt{2} - 1$ (dashed vertical line). Panel (d) magnifies this regime where ED data for the largest chains (symbols $L = 512, \ldots, 16384$) are compared to the analytical expression Eq. (3.7) (full line) with $\ell'_{\text{eff}}$ given by Eq. (3.3) and $\ell_{\text{eff}}$ by Eq. (3.9). Panel (e) tests the validity of the ansatz [27] Eq. (3.9) for $\ell_{\text{eff}}$ using two estimates: The lowest energy gap $\Delta_0(L)$ Eq. (3.5) and the surface magnetization $m^2(L)$ Eq. (3.6). The agreement bewteeen ED data (symbols) and the analytical expression Eq. (3.9) (line) gets clearly better when $L$ grows.

## IV.   THE ZERO-MODE OCCUPATION

### A.   Simple expectation in the topological regime

The MZM occupation number for an eigenstate $|n_p\rangle$ is

$$\mathcal{N}_{\text{MZM}}^{(n_p)} = \langle \Psi^\dagger \Psi \rangle_{n_p} = \frac{1}{2}\left(1 + i\langle \psi_a \psi_b \rangle_{n_p}\right), \qquad (4.1)$$

In the topological regime, the parity fidelity is $\mathcal{F}_{\text{MZM}} = 1$ and therefore Eq. (2.2) becomes

$$|n_{-p}\rangle = \frac{1}{2}\left(\psi_a + i\mathcal{S}_n\mathbb{P}\psi_b\right)|n_p\rangle, \qquad (4.2)$$

where $\mathcal{S}_n = \pm 1$ encodes the sign of the correlation between the boundary spins in the state $|n_p\rangle$. It is then straightforward to show that for any eigenstate $|n_p\rangle$ the MZM occupation

$$\mathcal{N}_{\text{MZM}}^{(n_p)} = \frac{1}{2}\left(1 - p\mathcal{S}_n\right). \qquad (4.3)$$

The 4 possible cases are summarized in Table I.

| Parity $p$ | End-to-end correlation $\mathcal{S}$ | MZM occupation $\mathcal{N}_{\text{MZM}}$ |
|---|---|---|
| +1 | −1 (AF) | 1 |
| −1 | +1 (FM) | 1 |
| +1 | +1 (FM) | 0 |
| −1 | −1 (AF) | 0 |

TABLE I. The occupation $\mathcal{N}_{\text{MZM}}^{(n_p)} = 0$ or $1$ in the topological regime, depends on the parity $p$ and the sign of edge spins $\mathcal{S}_n = \text{sgn}\left(C_{xx}^{\text{end}}\right)$.

A simple example is the ferromagnetic TFI chain which, in the limit of large coupling and small field limit $\Gamma = h/J \ll 1$, displays eigenstates with cat-states forms (in the $\{\sigma^x\}$ basis)

$$|n_p^{(\text{FM})}\rangle \approx \frac{|\uparrow\uparrow\downarrow\uparrow\uparrow \cdots \downarrow\uparrow\rangle + p|\downarrow\downarrow\uparrow\downarrow\downarrow \cdots \uparrow\downarrow\rangle}{\sqrt{2}} \qquad (4.4)$$

$$|n_p^{(\text{AF})}\rangle \approx \frac{|\uparrow\uparrow\downarrow\uparrow\uparrow \cdots \downarrow\downarrow\rangle + p|\downarrow\downarrow\uparrow\downarrow\downarrow \cdots \uparrow\uparrow\rangle}{\sqrt{2}}, \qquad (4.5)$$

where one sees that the sole difference concerns the edge spin correlations: ferromagnetic (FM) or antiferromagnetic (AF). Since the MZMs are essentially localized on the boundary operators $\psi_a \approx a_1 = \sigma_1^x$ and $\psi_b \approx b_L = -i\mathbb{P}\sigma_L^x$, one can rewrite the MZM occupation Eq. (4.1) as follows

$$\mathcal{N}_{\text{MZM}}^{(n_p)} \approx \frac{1}{2}\left(1 - \langle\mathbb{P}\sigma_1^x\sigma_L^x\rangle_{n_p}\right), \qquad (4.6)$$

which nicely matches Eq. (4.3) for the above cat-state situation.

However, in the general case the MZM correlator $i\langle\psi_a\psi_b\rangle_{n_p}$ is not a simple object to directly evaluate, except very deep in the topological regime where it reduces to the end-to-end correlation, which for cat-states is $\langle\sigma_1^x\sigma_L^x\rangle_{n_p} = \pm 1$. Away from the $\Gamma \to 0$ limit, it is quite interesting to notice that despite its much more complicated non-local form

$$i\langle\psi_a\psi_b\rangle_{n_p} = \frac{i}{N_a N_b}\sum_{j,j'}\Theta_j^a\Theta_{j'}^b\langle a_j b_{L+1-j'}\rangle_{n_p}, \quad (4.7)$$

it is expected to be $i\langle\psi_a\psi_b\rangle_{n_p} = \pm 1$. This will no longer be true in the non-topological regime where the norms $N_{a,b}$ diverge, and therefore $\langle\psi_a\psi_b\rangle_{n_p} \to 0$, yielding $\mathcal{N}_{\text{MZM}} \to 1/2$, as observed in Fig. 1 (b).

### B. Free fermion formulation

The non-interacting Kitaev chain Hamiltonian is easily diagonalized by a Bogoliubov transformation (see Appendix), and takes the following quadratic form

$$\mathcal{H}_K = 2\sum_{m=1}^{L}\epsilon_m\left(\phi_m^\dagger\phi_m - \frac{1}{2}\right), \qquad (4.8)$$

where $\phi_m$ are new fermionic modes, and the single particle energies are such that $0 \leq \epsilon_1 \leq \epsilon_2 \leq \ldots \epsilon_L$. In the topological regime, for large enough chain length $L$ one expects $\mathcal{F}_{\text{MZM}} = 1$, which implies that $\Psi^\dagger = \phi_1^\dagger$. This will essentially be true for system sizes much larger that the correlation length $\xi_{\text{zm}}$ (i.e. the localization length of the MZM). However, when $\xi_{\text{zm}}$ is not small as compared to the chain length $L$, the situation becomes quite interesting. Indeed, in this case the fidelity being different from unity, it is convenient to rewrite $\Psi^\dagger$ as

$$\Psi^\dagger = \mathcal{F}_{\text{MZM}}\phi_1^\dagger$$
$$+ \sum_{m\geq 2}\left(\frac{\mathcal{A}_m + \mathcal{B}_m}{2}\right)\phi_m^\dagger + \left(\frac{\mathcal{A}_m - \mathcal{B}_m}{2}\right)\phi_m, \quad (4.9)$$

where $\mathcal{A}_m$ and $\mathcal{B}_m$ depends on both the MZM coefficients $\Theta_j^{a,b}$ and the Bogoliubov transformation (see Appendix). This allows us to express the MZM occupation number as follows

$$\mathcal{N}_{\text{MZM}}^{(n_p)} = \mathcal{F}_{\text{MZM}}^2\langle\phi_1^\dagger\phi_1\rangle_{n_p} \qquad (4.10)$$
$$+ \sum_{m\geq 2}\left(\frac{1}{4}[\mathcal{A}_m - \mathcal{B}_m]^2 + \mathcal{A}_m\mathcal{B}_m\langle\phi_m^\dagger\phi_m\rangle_{n_p}\right).$$

Hence, since many-body eigenpairs $\{|n_p\rangle; |n_{-p}\rangle\}$ only differ by their occupation of the lowest single-particle mode $m = 1$, the difference in their MZM occupation is simply given by

$$\delta\mathcal{N}_{\text{MZM}}^{(n_p)} = \langle n_p|\Psi^\dagger\Psi|n_p\rangle - \langle n_{-p}|\Psi^\dagger\Psi|n_{-p}\rangle \quad (4.11)$$
$$= -p\mathcal{S}_n\mathcal{F}_{\text{MZM}}^2, \qquad (4.12)$$

where the sign prefactor depends on both the parity $p$ and the the sign $\mathcal{S}_n$ of the end-to-end spin correlation. This result is generally true across the entire phase diagram (even in the trivial regime, but only for finite $L$ because otherwise the strong MZM operator is not defined anymore).

This can be numerically checked, see Fig. 4 where the MZM occupations have been computed for random high-energy eigenstates, with or without the $m = 1$ mode occupied. While $\mathcal{N}_{\text{MZM}} \to 0$ or 1 in the topological regime for all energies, results are much more spread in the trivial regime where $\mathcal{N}_{\text{MZM}} \to 1/2$ slowly with increasing $L$. We also nicely check in Fig. 4 (c) that the difference Eq. (4.11) does not depend on the energy and is given by the square of the fidelity Eq. (4.12).

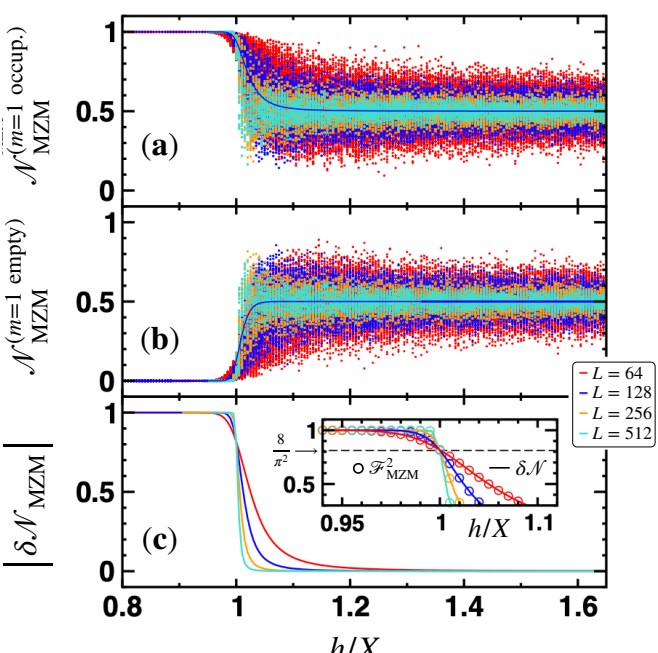

FIG. 4. ED results for the MZM occupation number $\mathcal{N}_{\text{MZM}}$ computed for the TFI model ($Y = 0$) against $h/X$, shown for 4 different system sizes. All symbols correspond to several hundreds of random (high-energy) eigenstates of the form $|n_p\rangle = \prod_{m=1}^{L}\Upsilon_m\phi_m^\dagger|\text{GS}\rangle$ where $\Upsilon_m = 0$ or 1 with probablity 1/2, with the lowest fermionic mode $m = 1$ being either (a) occupied or empty (b). The full blue line shows results for the two lowest energy states with $L = 128$ sites: (a) for the first excited state $|\text{FES}\rangle = \phi_1^\dagger|\text{GS}\rangle$, and (b) for the ground-state $|\text{GS}\rangle$. (c) The absolute value of the difference Eq. (4.11) is independent of the states, and the analytical prediction Eq. (4.12) is perfectly verified in the inset.

### C. Analytical derivation at criticality

Apart from the simple exact relation Eq. (4.12) which links the difference between MZM occupations and the fidelity, there is no easy way to analytically evaluate $\mathcal{N}_{\text{MZM}}$ for a generic eigenstate having the following form in the fermionic basis

$$| n_p \rangle = \prod_{m \text{ occupied}} \phi_m^\dagger | \text{GS} \rangle. \qquad (4.13)$$

Nevertheless, at criticality, using Eq. (4.1) and Eq. (4.7), it is straightforward to arrive for the TFI chain model at

$$\mathcal{N}_{\text{critical}} = \frac{1}{2}\Big(1 + i\langle \psi_a \psi_b \rangle\Big), \qquad (4.14)$$

$$= \frac{1}{2}\Big(1 + \frac{i}{L}\sum_{i=1}^{L}\sum_{j=1}^{L}\langle a_i b_j \rangle\Big) = \frac{1}{2} + \frac{1}{L}\sum_{ij}\langle c_i^\dagger c_j \rangle,$$

where we have used the simple form of the critical MZM at the Ising QCP of the TFI chain

$$\psi_a = \frac{1}{\sqrt{L}}\sum_{j=1}^{L} a_j \quad \text{and} \quad \psi_b = \frac{1}{\sqrt{L}}\sum_{j=1}^{L} b_j. \qquad (4.15)$$

It turns out that one can obtain a rather simple analytical expression for the above sum, building on the fact that an $L$-sites XY chain is equivalent to two decoupled Ising chains with $L/2$ sites [28, 29]. After a bit of manipulation we arrive at

$$\sum_{ij}\langle c_i^\dagger c_j \rangle \quad \approx \quad -\frac{2}{L}\sum_{j=1}^{L/2}\frac{j}{\sin\left(\frac{\pi j}{L}\right)} \qquad (4.16)$$

$$\xrightarrow[L\to\infty]{} \quad -\frac{2}{\pi^2}\int_0^{\pi/2}\frac{x}{\sin(x)}dx \qquad (4.17)$$

where one recognizes the Catalan's constant $\mathcal{G}$ [20], given by

$$\mathcal{G} = \frac{1}{2}\int_0^{\pi/2}\frac{x}{\sin(x)}dx \approx 0.915965594177\dots \qquad (4.18)$$

Therefore, the GS occupation of the MZM at criticality is expected to be

$$\mathcal{N}_{\text{critical}}^{(\text{GS})} = \frac{1}{2} - \frac{4}{\pi^2}\mathcal{G} \approx 0.128773127289\dots, \qquad (4.19)$$

for the GS, and for the first excited state $| \text{FES} \rangle = \phi_1^\dagger | \text{GS} \rangle$

$$\mathcal{N}_{\text{critical}}^{(\text{FES})} = \mathcal{N}_{\text{critical}}^{(\text{GS})} + \mathcal{F}_{\text{MZM}}^2 \approx 0.939342596428\dots \qquad (4.20)$$

### D. Finite size numerics

These analytical predictions for the critical MZM occupation numbers are numerically checked, both in the GS and the FES. Fig. 5 shows ED results for critical TFI chains, up to $L = 16384$ sites, where we nicely observe a finite-size convergence to the predicted values. Interestingly, as before with

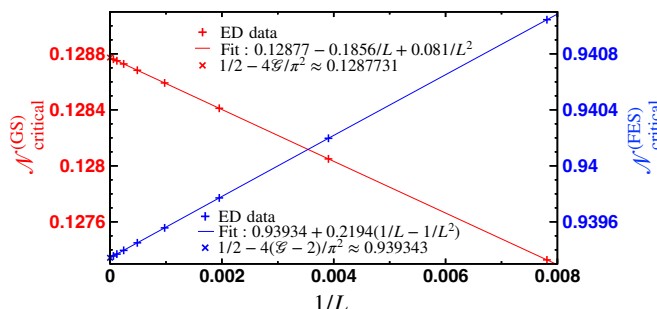

FIG. 5. Critical MZM occupation number $\mathcal{N}_{\text{MZM}}$ computed for the GS (red, left axis) and the FES (blue, right) of the TFI model ($Y = 0$) at criticality ($h = X$). Finite-size ED results (+) are perfectly fitted with a simple order 2 polynomial form as indicated on the plot (line); the inifinite-size extrapolations are also indicated on the graph (×).

the fidelity, here we also find along the critical line an exact vanishing of the finite-size corrections for the MZM occupations which occurs at the same special point: $\Delta/t = \sqrt{2} - 1$ in the Kitaev chain language (equivalently $Y/X = \sqrt{2} - 1$ and $h/X = \sqrt{2}$ for the XY spin chain).

In addition, we numerically observe a weak finite-size drift of this special point $Y^*(L)$ where the finite-size corrections vanish. This is shown in Fig. 6 where one sees the convergence of $Y^*(L)$ towards its asymptotic value $\sqrt{2} - 1$ with $L$ as a power-law $\sim 1/L$ for both quantities.

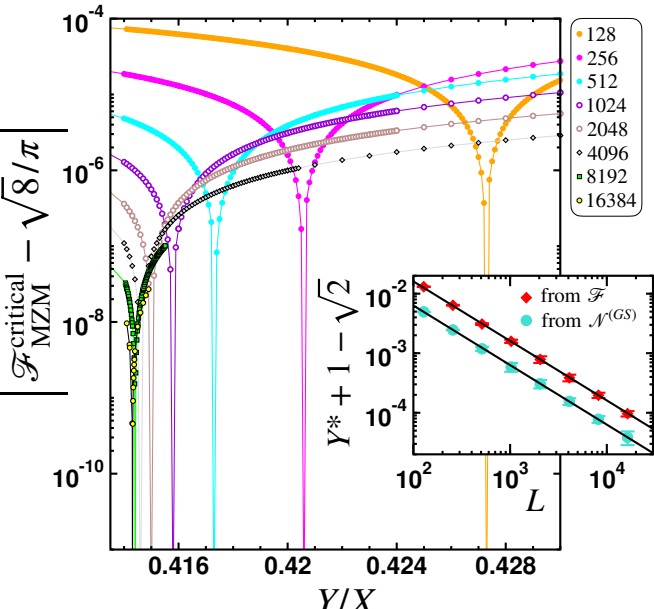

FIG. 6. Main panel: finite-size convergence of $\mathcal{F}_{\text{MZM}}^{\text{critical}}$ towards $\sqrt{8}/\pi$ along the critical Ising line. The vanishing of the finite-size corrections occurs at a length-dependent coupling $Y^*(L)/X$, which quickly converges to $\sqrt{2} - 1$ with increasing $L$. This convergence of $Y^*$ vs. $L$ ($X$ is set to 1) is reported in the Inset for both quantities $\mathcal{F}_{\text{MZM}}^{\text{critical}}$ and $\mathcal{N}_{\text{critical}}^{(GS)}$. Black lines are fits of the form $A/L$ with $A_{\mathcal{F}} \approx 1.608$ and $A_{\mathcal{N}} \approx 0.6317$. The convergence is even faster for $\mathcal{N}$.

## V. CONCLUSIONS AND DISCUSSIONS

### A. Summary of the main results

In this work, we have revisited the topological transition occurring in the paradigmatic Kitaev chain model. We introduced two topological markers to probe the Majorana zero mode (MZM) physics: the MZM fidelity $\mathcal{F}_{\mathrm{MZM}}$, see Eq. (1.9), and the MZM occupation number $\mathcal{N}_{\mathrm{MZM}}$, Eq. (1.8). Whereas these two quantities have long been known to take simple values in both trivial and topological regimes, we have analytically shown that both markers take a non-trivial universal value, constant along the quantum (1+1) Ising critical line of the XY-Kitaev chain model, as summarized in Tab. II.

The MZM fidelity, independent of the energy, has been analytically computed by establishing a connection with the surface magnetization $m^s$, the lowest-energy gap $\Delta_0$, and the norm of the commutator $C_\Psi = \left\| [\mathcal{H}_K, \Psi] \right\|$ between the Hamiltonian and the zero-energy Dirac fermion ($\Psi = (\psi_a + \mathrm{i}\psi_b)/2$ built from left and right MZMs), such that $\mathcal{F}_{\mathrm{MZM}} = m^s C_\Psi / \Delta_0$. These three quantities exhibit algebraic behavior at criticality, which compensates and gives rise to a universal value, that we strikingly find to remain constant along the critical Ising line: $\mathcal{F}_{\mathrm{MZM}}^{\mathrm{critical}} = \sqrt{8}/\pi$. We have also computed the finite-size corrections to this asymptotic result: we analytically identified a singular point along the Ising critical line where such corrections vanish completely for the special value $\Delta/t = \sqrt{2} - 1$ of the ratio between pairing and hopping in the Kitaev chain. These predictions were successfully compared with exact diagonalization results (up to several thousand lattice sites).

Concerning the MZM occupation number $\mathcal{N}_{\mathrm{MZM}}$ we have obtained exact results, universal along the Ising critical line, for two first energy levels, see Tab. II, and also for the zero-mode occupation difference $\delta\mathcal{N}_{\mathrm{MZM}}^{(n_P)}$ between all many-body eigenpairs $\left\{ |\, n_P \,\rangle \,; |\, n_{-P} \,\rangle \right\}$ that is given by $\mathcal{F}_{\mathrm{MZM}}^2$, see Eq. (4.12).

|  | Trivial | Critical | Topological |
|---|---|---|---|
| $\mathcal{F}_{\mathrm{MZM}}$ | 0 | $\frac{\sqrt{8}}{\pi} \approx 0.9003163$ | 1 |
| $\mathcal{N}_{\mathrm{MZM}}^{(\mathrm{GS})}$ | 0.5 | $\frac{1}{2} - \frac{4\mathcal{G}}{\pi^2} \approx 0.1287731$ | 0 |
| $\mathcal{N}_{\mathrm{MZM}}^{(\mathrm{FES})}$ | 0.5 | $\frac{1}{2} + \frac{8-4\mathcal{G}}{\pi^2} \approx 0.9393426$ | 1 |

TABLE II. Summary of the values taken by the two topological markers studied in this paper: the MZM fidelity $\mathcal{F}_{\mathrm{MZM}}$, see Eq. (1.9), and the MZM occupation number $\mathcal{N}_{\mathrm{MZM}}$, Eq. (1.8), for the ground-state (GS) and the first excited state (FES). They all take non-trivial values at the Ising quantum critical point between trivial and topological regimes, values that are universal for (1+1) Ising criticality.

### B. Open questions and possible future directions

The rather simple analytical expression of the MZM operators Eq. (1.4) has clearly helped us to derive exact results at criticality. In addition, the free-fermion nature allowed to numerically verify our analytical predictions with great accuracy using very large Kitaev chains, up to $L = 16384$ lattice sites. Nevertheless, there are still many open questions and directions for which we can imagine several extensions to go beyond the case of non-interacting clean Kitaev chains.

A first natural development concerns the effects of the environment, such as in open systems described by non-Hermitian models [30, 31], or when quenched disorder is added directly to the Kitaev Hamiltonian [32–40]. This last case has a particularly long and dense history, especially for the magnetic version of the problem, going back to the seminal work of D. S. Fisher [41, 42]. Subsequent progress has been made [43–52], giving a fairly complete description of the non-interacting random problem, and especially of the very unusual properties of the infinite randomness criticality fixed point (IRFP) [28, 42]. It would therefore be quite interesting and relevant to revisit the Kitaev chain model in the presence of quenched disorder using the topological markers introduced in the present paper, extending the results obtained at the clean (1+1) Ising critical point to the physics of the IRFP.

Another very important ingredient concerns the effect of interactions on MZMs [53–57]. In particular it is known that the construction of MZM operators is a very difficult task in the general interacting case [12–14, 58–61]. However, a distinction should be made between the integrable case, e.g. for the XYZ model where an exact construction has been shown possible in the clean case [12], and the non-integrable case, e.g. provided by the interacting Ising-Majorana chain model [13, 14, 61–65] where the MZMs are only *almost strong* [13, 14, 61, 66]. Nevertheless, it would be very interesting to consider the possibility of checking in one way or another the universality of our results against finite interaction along the self-dual Ising critical line of the interacting Majorana chain model [18, 67].

A clearly challenging direction touches the combined effects of disorder and interactions which brings a full set of very exciting questions, some of them being related to the celebrated many-body localization (MBL) problem [68, 69]. For instance it was recently found using large-scale DMRG simulations that interactions are not relevant to the IRFP of the random TFI chain model at zero temperature [70], while Monthus had previously shown [71] that a strong-disorder RG treatment generates higher-order terms that prevent a conclusion, in contrast to the XXZ case [28]. The disordered XYZ chain was also shown to display very rich physics both at zero temperature [72], and at high energy [73, 74]. In such a context, the possible existence and stability of MZMs and their fate at criticality in the presence of both disorder and interactions remains a fascinating subject, as it has been little discussed and has shown contrasting conclusions [75–80].

Finally, it is also worth mentioning the very interesting case of driven systems [81–83] which has strong experimental relevance [5, 84–86].

**ACKNOWLEDGMENTS**

I would like to express my sincere thanks to Natalia Chepiga for collaborating on related matters. I also acknowledge some discussions with Paul Fendley and Jack Kemp. This work was granted access to the HPC resources of CALMIP center under the allocation 2022-P0677 as well as GENCI (grant x2021050225).

## Appendix A: Additional free-fermion results

### 1. Exact diagonalization of the Kitaev model

In the general case where translational invariance is absent, we use the Nambu formalism [43] in order to solve the (not necessarily homogeneous) free-fermion Kiteav chain Hamiltonian Eq. (1.1). Introducing $\varphi^\dagger = \left( c_1^\dagger \cdots c_L^\dagger \; c_1 \cdots c_L \right)$, we arrive at the simple matrix representation of the Hamiltonian

$$\mathcal{H}_{\mathrm{K}} = \varphi^\dagger \begin{pmatrix} M & \widetilde{M} \\ -\widetilde{M} & -M \end{pmatrix} \varphi. \tag{A1}$$

$M$ and $\widetilde{M}$ are $L \times L$ matrices, such that

$$M_{j,j} = \frac{\mu_j}{2} = h_j,$$
$$M_{j,j+1} = M_{j+1,j} = -\frac{t_j}{2} = -\frac{X_j + Y_j}{2}, \tag{A2}$$

while $M_{i,j} = 0$ elsewhere (unless periodic boundary conditions are used: $M_{1,L} = M_{L,1} = pt_L/2$, where $p = \pm 1$ is the fermionic parity). The matrix $\widetilde{M}$ instead is anti-symmetric:

$$\widetilde{M}_{j,j+1} = -\widetilde{M}_{j+1,j} = -\frac{\Delta_j}{2} = -\frac{X_j - Y_j}{2}, \tag{A3}$$

and $\widetilde{M}_{i,j} = 0$ otherwise. The free-fermion problem can be solved by diagonalizing the $2L \times 2L$ Hamiltonian matrix Eq. (A1), and we then rewrite the quadratic Hamiltonian

$$\mathcal{H}_{\mathrm{K}} = 2 \sum_{m=1}^{L} \epsilon_m \left( \phi_m^\dagger \phi_m - \frac{1}{2} \right), \tag{A4}$$

with single particle energies $0 \leq \epsilon_1 \leq \epsilon_2 \leq \ldots \epsilon_L$.

### 2. Majorana zero mode operators and fidelity

The new fermionic modes are given by the Bogoliubov transformation $\phi_m^\dagger = \sum_{i=j}^{L} \left( u_j^m c_j^\dagger + v_j^m c_j \right)$, with real $u_j^m$ and $v_j^m$. The inverse transformation is $c_j^\dagger = \sum_{m=1}^{L} \left( u_j^m \phi_m^\dagger + v_j^m \phi_m \right)$,

from which one can express the MZM operators Eq. (1.4)

$$\psi_a = \frac{1}{N_a} \sum_{j=1}^{L} \Theta_j^a a_j = \sum_{m=1}^{L} \mathcal{A}_m \left( \phi_m^\dagger + \phi_m \right) \tag{A5}$$

$$\psi_b = \frac{1}{N_b} \sum_{j=1}^{L} \Theta_j^b b_{L+1-j} = \mathrm{i} \sum_{m=1}^{L} \mathcal{B}_m \left( \phi_m^\dagger - \phi_m \right), \tag{A6}$$

where

$$\mathcal{A}_m = \frac{1}{N_a} \sum_{j=1}^{L} \Theta_j^a \left( u_j^m + v_j^m \right) \tag{A7}$$

$$\text{and} \quad \mathcal{B}_m = \frac{1}{N_b} \sum_{j=1}^{L} \Theta_j^b \left( u_{L+1-j}^m - v_{L+1-j}^m \right). \tag{A8}$$

The zero-energy Dirac fermion creation operator Eq. (1.7) is therefore given by

$$\Psi^\dagger = \sum_{m=1}^{L} \left( \frac{\mathcal{A}_m + \mathcal{B}_m}{2} \right) \phi_m^\dagger + \left( \frac{\mathcal{A}_m - \mathcal{B}_m}{2} \right) \phi_m, \tag{A9}$$

from what we can simply express the MZM fidelity. Indeed, since many-body eigenpairs only differ by their occupation of the lowest single-particle mode $m = 1$ one has

$$\mathcal{F}_{\mathrm{MZM}} = \frac{\mathcal{A}_1 + \mathcal{B}_1}{2}, \tag{A10}$$

which further simplifies in the clean case where left and right boundaries are equivalent, yielding

$$\mathcal{F}_{\mathrm{MZM}} = \frac{1}{N_0} \sum_{j=1}^{L} \Theta_j^a \left( u_j^1 + v_j^1 \right). \tag{A11}$$

## Appendix B: Surface magnetization

### 1. Definition

The surface magnetization [23, 24] is defined by

$$m_{1,L}^s = \langle n_- | \sigma_{1,L}^x | n_+ \rangle. \tag{B1}$$

The boundary magnetization can be expressed as follows

$$\sigma_1^x = a_1 = c_1^\dagger + c_1$$
$$= \sum_{m=1}^{L} \left( u_1^m + v_1^m \right) \left( \phi_m^\dagger + \phi_m \right), \tag{B2}$$

and

$$\sigma_L^x = \mathrm{i}\mathbb{P} b_L = -\mathbb{P} \left( c_L^\dagger - c_L \right)$$
$$= -\mathbb{P} \sum_{m=1}^{L} \left( u_L^m - v_L^m \right) \left( \phi_m^\dagger - \phi_m \right). \tag{B3}$$

### 2. Invariance across the many-body spectrum

Since two partner states $|n_p\rangle$ and $|n_{-p}\rangle$ only differ by their occupation of the lowest single-particle mode $m = 1$, one gets

$$m_1^s = u_1^1 + v_1^1, \tag{B4}$$

which we take positive by convention. On the other hand, at the right boundary we have

$$m_L^s = \left(u_L^1 - v_L^1\right) \langle n_- | \left(\phi_m^\dagger - \phi_m\right) | n_+\rangle, \tag{B5}$$

whose sign depends on the state. Indeed, if the even ($p = +1$) many-body eigenstate $|n_+\rangle$ has the lowest single-particle mode $m = 1$ which is empty (like the GS for instance), we have $m_L^s = u_L^1 - v_L^1$ ($= m_1^s$ if the system is clean). Conversely if the mode $m = 1$ is occupied, one gets $m_L^s = v_L^1 - u_L^1$ ($= -m_1^s$ again if the system is clean). We therefore see that the magnitude of the surface magnetization does not depend on the many-body energy $E_n^\pm$, but only on the coeffiient of the lowest ($m = 1$) single particle mode. Only its relative sign between left and right boundaries oscillates across the spectrum, see also Tab. I.

### 3. Critical scaling of the surface magnetization

In the ordered regime, for $L \gg \xi_{zm}$, one can use the mapping $|n\rangle_+ = \psi_a |n\rangle_-$, that brings us to the following relation

$$m_1^s = \langle n_- |\sigma_1^x \psi_a| n_-\rangle = \frac{1}{N_a}, \tag{B6}$$

where the MZM norm results from a geometrical sum of exponentials, see Eq. (1.5) and Eq. (1.6), simply yielding

$$\frac{1}{N_a} = \sqrt{1 - \exp(-2/\xi_{zm})}. \tag{B7}$$

When approaching the critical point from the ordered regime, the length scale $\xi_{zm}$ becomes very large, and therefore the surface magnetization is given by

$$m_1^s \approx \sqrt{2/\xi_{zm}}. \tag{B8}$$

When $\xi_{zm} \gg L$, it is rather standard to replace this length scale by the lattice size $L$ in Eq. (B8). In the main text, we have argued that the correct length scale to use is the one which also enters in the wave vector Eq. (2.13) $k_{min}$, i.e. $L+\ell_{eff} = L+1/2$.

Fig. 7 provides the numerical justification for this: ED data for $m_1^s(L)$ are shown against $1/L$ up to $L = 10^4$ for the transverse field Ising chain at criticality. A comparison with the analytical form $\sqrt{\frac{2}{L+\ell_{eff}}}$ is shown for $\ell_{eff} = 0$ (open black circles) and $\ell_{eff} = 1/2$ (open red squares). It is clearly visible that using a finite shift $\ell_{eff} = 1/2$ gives a much better description (see inset).

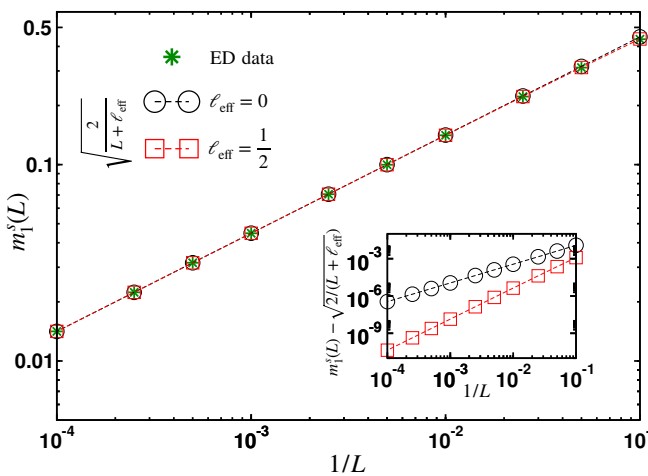

FIG. 7. Critical scaling of the surface magnetization for the TFI chain at $\Gamma = h/X = 1$. ED data are compared with the expected algebraic behavior $\sqrt{2/(L + \ell_{eff})}$.

### Appendix C: MZM in the general case of the XY-Kitaev chain

#### 1. Analytical expression in the ordered regime

##### a. Iterative procedure

Using Majoranas, the clean XY-Kitaev model reads

$$\mathcal{H}_{XY-K} = i \sum_j \left(X b_j a_{j+1} - Y a_j b_{j+1} + h a_j b_j\right). \tag{C1}$$

Assuming simple linear combinations for the MZM operators

$$\Psi_a = \frac{1}{N_a} \sum_{j=1}^{L} \Theta_j \, a_j \quad \text{and} \quad \Psi_b = \frac{1}{N_b} \sum_{j=1}^{L} \Theta_j \, b_{L+1-j}, \tag{C2}$$

and using $[\mathcal{H}_{XY-K}, a_j] = 2i\left(X b_{j-1} - h b_j + Y b_{j+1}\right)$ and $[\mathcal{H}_{XY-K}, b_j] = -2i\left(Y a_{j-1} - h a_j + X a_{j+1}\right)$, we iteratively arrive at the simple recursion relation for $\Theta_j = \Theta_j^{a,b}$:

$$\Theta_{j+1} = \frac{h}{X}\Theta_j - \frac{Y}{X}\Theta_{j-1}, \tag{C3}$$

such that

$$[\mathcal{H}_{XY-K}, \Psi_a] = 2i\frac{1}{N_a}\left(Y\Theta_{L-1} - h\Theta_L\right) b_L \tag{C4}$$

$$[\mathcal{H}_{XY-K}, \Psi_b] = -2i\frac{1}{N_b}\left(Y\Theta_{L-1} - h\Theta_L\right) a_1. \tag{C5}$$

One solve the recursion Eq. (C3) with initial conditions $\Theta_0 = 0$ and $\Theta_1 = 1$, restricting to positive couplings $h$, $X$, $Y \geq 0$, and $X \geq Y$ (other cases can be easily derived).

##### b. MZM

We note in passing that the phase diagram of the XY-Kitaev chain model, shown in Fig. 1 (a), can simply be inferred from

the existence of normalizable MZMs, which requires that the largest eigenvalue of the Eq. (C3) has its modulus less that one, i.e. if $X + Y > h$. Contrary to the TFIM case, here the topological regime is richer as one can distinguish two types of MZM decays, incommensurate and commensurate.

*(i) Incommensurate regime ($h^2 < 4XY$).*

In this case,

$$\Theta_j = \frac{2X}{\sqrt{4XY - h^2}} \sin(\varphi j) \, e^{-j/\xi_{zm}} \qquad (C6)$$

displays oscillations and exponential decay, controlled by

$$\cos\varphi = \frac{h}{2\sqrt{XY}} \quad \text{and} \quad \frac{1}{\xi_{zm}} = \ln\sqrt{\frac{X}{Y}}. \qquad (C7)$$

The MZM normalization factor $N_a = N_b \equiv N_0$ can be evaluated in the large $L$ limit

$$\frac{1}{N_0} \xrightarrow[L\to\infty]{} \sqrt{2} \sin\varphi \left[ \frac{1}{1 - \frac{Y}{X}} + \frac{\frac{Y}{X} - \cos(2\varphi)}{1 - 2\frac{Y}{X}\cos(2\varphi) + \left(\frac{Y}{X}\right)^2} \right]^{-\frac{1}{2}}. \qquad (C8)$$

*(ii) Commensurate regime ($h^2 \geq 4XY$).* In this case

$$\Theta_j = \frac{X}{\alpha h} \left(\frac{h}{2X}\right)^j \times \left[ (1+\alpha)^j - (1-\alpha)^j \right]$$
$$\xrightarrow[j\gg 1]{} \begin{cases} \frac{X}{\alpha h} e^{-j/\xi_{zm}} & \text{if } 2\sqrt{XY} > h > X + Y \\ \infty & \text{if } h > X + Y, \end{cases} \qquad (C9)$$

where the edge mode localization length $\xi_{zm}$ is given by

$$\frac{1}{\xi_{zm}} = \ln\left[ \frac{2X}{(1+\alpha)h} \right], \qquad (C10)$$

and $\alpha = \sqrt{1 - 4XY/h^2}$. The MZM normalization factor can

be expressed in the large chain $L$ limit:

$$\frac{1}{N_0} \xrightarrow[L\to\infty]{} 2\alpha \left[ \frac{1}{(1+\alpha)^{-2} - \left(\frac{h}{2X}\right)^2} + \frac{1}{(1-\alpha)^{-2} - \left(\frac{h}{2X}\right)^2} \right.$$
$$\left. - \frac{2}{(1-\alpha^2)^{-1} - \left(\frac{h}{2X}\right)^2} \right]^{-\frac{1}{2}}, \quad \text{if } \alpha \neq 0. \qquad (C11)$$

When $Y = 0$ ($\alpha = 0$), we simply recover the TFIM result Eq. (2.7), i.e. $1/N_0 \xrightarrow[L\to\infty]{} \sqrt{1 - h^2/X^2}$.

### 2. Critical behavior

*(i) Critical commutators.* At criticality when $h = X + Y$, the MZM coefficients do not decay anymore and are given by

$$\Theta_j^{critical} = \frac{X}{X - Y} \left[ 1 - \left(\frac{Y}{X}\right)^j \right]. \qquad (C12)$$

Therefore, the critical commutators are easy to compute: Eq. (C5) thus becomes

$$[\mathcal{H}_{XY-K}, \Psi_a] = -2i\frac{X^2}{N_a(X-Y)} \left[ 1 - \left(\frac{Y}{X}\right)^{L+1} \right] b_L$$

$$[\mathcal{H}_{XY-K}, \Psi_b] = 2i\frac{X^2}{N_b(X-Y)} \left[ 1 - \left(\frac{Y}{X}\right)^{L+1} \right] a_1, \quad (C13)$$

which indeed rapidly converges ($Y < X$) at large enough $L$ to the results Eq. (3.1) that we rewrite here

$$[\mathcal{H}_{XY-K}, \psi_a] = \frac{-2iX^2}{N_a(X-Y)} b_L \qquad (C14)$$

$$[\mathcal{H}_{XY-K}, \psi_b] = \frac{2iX^2}{N_b(X-Y)} a_1. \qquad (C15)$$

*(ii) Critical MZM norm.* Building on the critical coefficients Eq. (C12), one easily gets the norm (using $y = Y/X$)

$$N_{a,b} = N_0 = \sqrt{\sum_{j=1}^{L} \left(\Theta_j^{critical}\right)^2}$$
$$= \frac{X}{X-Y} \sqrt{L - \frac{y(2+y)}{1-y^2}} \qquad (C16)$$

thus yielding Eq. (3.2) with the effective length shift $\ell'_{eff}$ given by Eq. (3.3).

---

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
