# Peer review of "Universal signatures of Majorana zero modes in critical Kitaev chains"

_SciPost Physics_

## Round 1 · Referee Report · Anonymous (Referee 1) · 2024-4-7

Report

This Manuscript tackles some intriguing and apparently overseen aspects of an otherwise vastly investigated model, namely the Kitaev-Majorana chain. The presentation is well-organized, with clear — almost pedagogical — definition of the notation in most places, complemented by compact and easily accessible appendices with plenty of useful formulas and derivations.

The main result is the analysis of two topological markers (fidelity and occupation number) to probe the presence of Majorana zero modes (MZM): while they have been previously known to take simple values in the trivial and topological regimes, here the Author shows that they also take a non-trivial universal value, constant along the critical line separating the two phases. Personally, I would have considered the relation between these two (see Eq. (4.12)) as a noticeable result, too.

As a matter of taste, it could have been better to have a stronger introduction with the red-line of the whole story, to be then deepened in the bulk of the manuscript, rather than delving immediately in the notation details and spreading results all over the place. Anyway, this is no major criticism.

A number of interesting and relevant open questions are listed at the end, though some are not fully clear to me: e.g., in which sense would the results presented here help to clarify the long-standing question of the possible existence and stability of MZMs and their fate at criticality in the presence of both disorder and interactions. It would be great if the Author could provide a couple more insights about the envisioned roadmap.

Overall, up to minor questions and clarifications listed below, the Manuscript certainly presents a solid study, contains enough message and it is written in a suitable way to deserve publication on SciPost Core Physics.

Incidentally, I sincerely apologise for the delay incurred in preparing this Report, which was not in any way related to the quality of the Manuscript.

—————

1) In some places, the two topological markers examined here are dubbed as “introduced”, while in others they are acknowledged as previously known and simply overlooked… A streamline of the nomenclature would help the readership to correctly assess the originality of the study.

2) In Fig.1, panel c, the MZM fidelity of Eq. (1.9) is shown: however, the latter has an explicit dependence on the index n, which does not appear in the figure. Only much later in the text, it is claimed that \mathcal{F}^{(n)} is constant \forall n. It would be great if a comment in that sense is anticipated here, to avoid confusion in the readership.

3) In Eq. (2.2) what is \alpha_n? is this related to the \Theta coefficients in Eq. (1.4), or is that just a placeholder for a generic linear combination? Sorry, if I overlooked an explanation somewhere.

4) In the same Eq. (2.2), what is the need for the parity string, which was absent instead in Eq. (1.7)? More generally, I feel there is a bit too much jumping back and forth between the fermionic and the spin language (related by Jordan-Wigner transformation, indeed). It might be possible to ease the reading flow.

5) Minor: just before Eq. (2.7), there is a reference to A+, which is probably one of the appendices, but it is undefined…

6) In Eq.(2.8) the lowest energy gap \Delta_0 appears, while inverting Eq.(2.6) would have shown a relation to the parity gap and a dependence on the state n, which suddenly disappear. There might be something obvious that I am overlooking here, but this touches exactly upon the same point 2) mentioned above. One short explanation would solve both.

7) Sec. II.B.1 signals quite a jump with respect to the pedagogical character of the text up to that point: results and formulas are thrown into the game by picking them up from references, and not by re-deriving them -- perfectly fine, per se, but quite a change in style. Can the Author smoothen the step?

8) Is a more explicit relation between Eq. (3.4) and (2.8) available?

9) In Sec. III.A.2, is there a particular meaning in Eq. (3.9) beyond the cancellation of finite-size corrections at a specific value of the parameters? What I mean is whether there is any physical interpretation of those values: a comment would be appreciated. By the way, in the text just above the equation, it is mentioned that “it is more cumbersome to explicitly compute \ell^’{\rm{eff}}” —> should it not be \ell, since the one with ^’ is known from Eq.(3.3)?}

10) Minor: In Fig. 3, should the labels inside the panels a & b not be Y/X=0/.5 and h/X = 1.2 (& 1.6)? Incidentally, font sizes are a bit small inside the plots (axes labels are fully ok, instead).

11) If I am not mistaken, the statement below Eq. (4.1) about the parity fidelity being 1 in the topological regime strictly holds only in the thermodynamic regime (away from sweet parameter points, at finite sizes corrections are expected, though weak). Can the Author clarify it?

12) The notation in Eqs. (4.4) and (4.5) is not fully clear to me: what is the relation between the level index n and the number of "flipped" spins? It might be worth to take a couple more lines to introduce a careful notation (as done in most of the Manuscript). Similarly for the derivation of Eq. (4.7)…

13) As hinted in the general comment above, Eq. (4.12) relating the difference in the MZM occupation numbers and the fidelity marker (via the parity p and the the sign S𝑛 of the end-to-end spin correlation) seems to me a major result that would deserve more highlighting from the very beginning.

14) In Sec. IV.C there is an important statement about “an L-sites XY chain” being “equivalent to two decoupled Ising chains with L/2 sites”. It would be nice to have a sketchy reminder of that in an appendix, for the sake of keeping the work self-contained. The subsequent Eqs. (4.16)-(4.17) are again an example of a marked jump in the presentation style, which can be smoothened, especially given the absence of length constraints.

15) In App. A1, it seems that the construction works independently of the specific values of the coefficients \Theta_j^{a,b}. If this is the case, it would be great to provide a short explanation to the readership, in the same pedagogical spirit in which most of the work is written.

16) It is most probably due to some glitch in the BibTeX style, and it is matter for proofs, but some references like [5] and [6] do read a bit weird…

---

## Round 1 · Referee Report · Dirk Schuricht (Referee 2) · 2025-4-30

Report

The work presents analytical results for the Majorana zero mode fidelity and occupation number in the XY chain. The results are obtained analytically and checked against large scale numerical simulations. Somewhat surprisingly universal results are found along the Ising transition. The discussion of the open questions is very knowledgable and complete. Given the claim of universality I would like to see how the results behave when an interaction term in the form of the ANNNI chain is added.

The manuscript is clear and well written. I fully support publication in SciPost Physics Core.

Recommendation

Accept in alternative Journal (see Report)

---

## Editorial Decision

awaiting_resubmission